

# An efficient combined intelligent system for segmentation and classification of lung cancer computed tomography images

Maheswari Sivakumar[1], Sundar Chinnasamy[2] and Thanabal MS[3]

[1] Department of AI&DS, Dhirajlal Gandhi College of Technology, Salem, Tamilnadu, India
[2] Department of Computer Science and Engineering, Christian College of Engineering and Technology, Dindigul, India
[3] Department of Computer Science and Engineering, PSNA College of Engineering and Technology, Dindigul, India

Corresponding author
Maheswari Sivakumar,
maheswari.aids@dgct.ac.in

## ABSTRACT

**Background and Objective.** One of the illnesses with most significant mortality and morbidity rates worldwide is lung cancer. From CT images, automatic lung tumor segmentation is significantly essential. However, segmentation has several difficulties, such as different sizes, variable shapes, and complex surrounding tissues. Therefore, a novel enhanced combined intelligent system is presented to predict lung cancer in this research.

**Methods.** Non-small cell lung cancer should be recognized for detecting lung cancer. In the pre-processing stage, the noise in the CT images is eliminated by using an average filter and adaptive median filter, and histogram equalization is used to enhance the filtered images to enhance the lung image quality in the proposed model. The adapted deep belief network (ADBN) is used to segment the affected region with the help of network layers from the noise-removed lung CT image. Two cascaded RBMs are used for the segmentation process in the structure of ADBN, including Bernoulli–Bernoulli (BB) and Gaussian-Bernoulli (GB), and then relevant significant features are extracted. The hybrid spiral optimization intelligent-generalized rough set (SOI-GRS) approach is used to select compelling features of the CT image. Then, an optimized light gradient boosting machine (LightGBM) model using the Ensemble Harris hawk optimization (EHHO) algorithm is used for lung cancer classification.

**Results.** LUNA 16, the Kaggle Data Science Bowl (KDSB), the Cancer Imaging Archive (CIA), and local datasets are used to train and test the proposed approach. Python and several well-known modules, including TensorFlow and Scikit-Learn, are used for the extensive experiment analysis. The proposed research accurately spot people with lung cancer according to the results. The method produced the least classification error possible while maintaining 99.87% accuracy.

**Conclusion.** The integrated intelligent system (ADBN-Optimized LightGBM) gives the best results among all input prediction models, taking performance criteria into account and boosting the system's effectiveness, hence enabling better lung cancer patient diagnosis by physicians and radiologists.

## INTRODUCTION

Lung cancer is a tumor most commonly impacted by outside causes that typically influence the respiratory system (*Nanglia et al., 2020*). Compared to the 2005 survey, there were 159,292 more fatalities in 2018, an increase of 25% (*Chaunzwa et al., 2021*). Lung cancer will affect 234,030 persons in the United States in 2018, according to the North American Association of Central Cancer Registry's US report (*Joshua et al., 2021*). Additionally, according to a report by the American Cancer Society, there were 228,150 new lung cancer cases in the USA in 2019. 116,440 men and 111,710 women are impacted (*Murugesan et al., 2022*). According to the data, lung cancer caused the deaths of 142,670 persons (76,650 men and 66,020 women) (*Said et al., 2023*).

The survey's ultimate finding is that the proportion of persons affected by lung cancer has progressively climbed over the past five years (*Manoharan & Sathesh, 2020*). According to the data, doctors consider lung cancer the most prevalent condition when trying to diagnose it at an early stage (*Varchagall et al., 2022*). The symptoms of the disease are typically used to forecast lung cancer manually, such as change of sputum color, voice change, facial swelling, bleeding, neurological problems, headache, joint pains, bone fracture, memory loss, weight loss, shortage of breath, fatigue, chest pain, and blood coughing (*Ajai & Anitha, 2022*). Different screening techniques, including blood testing, biopsy, fluid biopsy, reflex testing, and genetic testing, have been employed continually for evaluation once the patient has been influenced by these technologies (*Hesamian et al., 2021*; *Sori et al., 2021*). For predicting lung cancer, lung cells and cell variations in the image are successfully examined by the proposed screening methodologies (*Dutande, Baid & Talbar, 2021*). One screening strategy is computerized tomography, which efficiently examines changes and deviations in the human body seen when exposed to X-rays (*Ahmed et al., 2020*). An automated system is developed to predict lung cancer from CT scans, and it uses numerous established techniques to do so, including image pre-processing, segmentation of disease region, extraction of cancer features, selection of related features, and lung cancer classification (*Xu, Wang & Razmjooy, 2022*). The significant functions in the previously described stage are the region segmentation and highlight selection procedure because the differences in the cancer image and the normal image are successfully identified by the successful prediction-affected region (*Asuntha & Srinivasan, 2020*).

Additionally, significant cancer features are extracted with the help of a segmented region, which lowers the system's complexity (*Chen et al., 2021*; *Almukhtar, 2023*). For cancer prediction, computation time and data over-fitting problems are reduced by using the feature selection process. The relevant regions from the acquired X-ray pictures are extracted using many segmentation approaches (*Gan et al., 2021*). Several features are extracted from the segmented region, and significant features are extracted using a variety of feature selection approaches (*Wu et al., 2020*). Several deep learning and machine

learning classifiers are used to categorize these selected features of lung cancer (*Liu, Zhao & Pang, 2020*). Despite effectively predicting lung cancer, typical automated systems still have problems with identification accuracy and take longer to analyze vast amounts of data (*He et al., 2022*). The method also needs to process the very minimum quality CT scans, which might result in false lung features and a higher probability of misclassification (*Maleki & Niaki, 2020*). The different authors then offered their opinions on the method for detecting lung cancer since their ideas led to the creation of an intelligent cancer prediction system. Despite the employment of various approaches, misclassification and large-scale data processing still need to improve.

Intelligent techniques are introduced in this research to enhance the process of detecting entire lung cancer process for resolving above-discussed problems. To remove image noise, average filter and adaptive median filter pre-process the acquired CT lung images, and histogram equalization is used to improve the lung image quality. The affected region is effectively extracted using ADBN due to the significance of the segmentation process. The hybrid SOI-GRS approach is used for selecting the optimized and effective features from the derived region, and then the optimized LightGBM classifier is used for lung tumor classification. Finally, a Python tool is used to create the required intelligent technique-based cancer detection system, and various performance indicators are used to assess the system's effectiveness.

## Major contribution

- The adaptive median and average filters are first used to pre-process the input CT images to remove noise. Then, histogram equalization is applied to the filtered images to improve lung image quality.
- The adapted deep belief network (ADBN) is utilized from the enhanced image to efficiently segment the affected region by employing many layers of the network after image pre-processing.
- The hybrid spiral optimization intelligent-generalized rough set (SOI-GRS) approach is used from the derived region for effective feature selection.
- After feature extraction, an optimized LightGBM model using the EHHO algorithm is used to classify lung cancer. Spiral settings and an approximation approach are used to select optimized features correctly.
- LUNA 16, KDSB, CIA, and the local dataset are used to train and test the proposed technique. Finally, a Python tool is used for developing a cancer diagnosis system based on an intelligent approach (ADBN-optimized LightGBM), and the system's effectiveness is assessed using various evaluation measures.

The remaining portions of the article are given as 'Related Prior Works' presents the literature reviews relevant to the proposed lung cancer classification system. 'Proposed Methodology' provides a proposed technique's thorough explanation. 'Experimental Results' provides the proposed technique's results and compares thoroughly to current approaches. Finally, 'Conclusion and Future Works' concludes by summarizing the entire process.

# RELATED PRIOR WORKS

Finding lung tumors is an important responsibility, and early identification of lung cancer is required to lower death rates and ensure that proper treatment is given. Lung cancer may be detected using a variety of computational algorithms, and multiple research approaches are discussed in this section. We have thus examined the methods for CT image-based segmentation and classification of lung cancer in the sections below.

For lung tumor segmentation, an M-SegSEUNet-CRF approach was developed by _Zhang et al. (2022)_ using CT images. The tumor regions are highlighted by embedding the SE blocks in 3D UNet using the spatially adaptive attention mechanism. The tumor borders are precisely defined by adding a dense connected CRF framework. A multi-scale approach addresses the issue of fluctuating tumor volume. Second, a SegSEUNet is trained using each set of image and mask cubes. The segmentation result is obtained by inputting each image cube into the trained SegSEUNet during testing, which appears as a 3D probability map. The tumor borders for each slice of the cube are accurately defined by combining the probability maps of four scales, and these combined maps are entered into the CRF model.

A novel automatic diagnostic categorization approach for CT images of the lungs was described by _Lakshmanaprabu et al. (2019)_. The ODNN-LDA was used to analyze this article's CT scan lung images. LDR is used to lower the dimensionality of the deep features collected from a CT lung image to classify lung nodules as benign or malignant. The Modified Gravitational Search Algorithm (MGSA) is used to optimize the ODNN after it has been applied to CT scans to classify lung cancer.

An effective DGMM-RBCNN model was investigated by _Jena, George & Ponraj (2021)_. Wiener and Gaussian filters are used to remove Gaussian noise. The pre-processing process reduces the noise in the images. A region-growing segmentation approach accomplishes an accurate ROI segmentation. The seed points are selected in a region-growing segmentation, and a larger region is obtained by merging the adjacent pixels in the pre-processed image. Then, essential features for a nodule of interest are retrieved. Using DGMM-RBCNN, the dimensionality is decreased from these derived features.

_Naseer et al. (2023)_ developed a deep-learning model for lobe segmentation and nodule identification of lung cancer. Three steps make up the examined classification model. Initially, it uses a CT slice to segment the lobe and a modified U-Net to predict the mask and to extract a candidate nodule; the second step uses a modified U-Net architecture while using a predicted mask and label. The support vector machine is used in the third phase based on modified AlexNet to categorize cancer nodules.

For the segmentation of lung tumors, a SegChaNet model was created by _Cifci (2022)_. From the surrounding chest region, the U-Net-CAM is used for effective segmentation. The trail of encoders used in the SegChaNet approach is used for feature extraction. From the encoded feature map collection, the multi-scale, dense-feature extraction model is developed for multi-scale feature extraction. By utilizing the decoders, we determined the lung segmentation map. The network remains invariant to the extent of the dense abnormality since the model has trained to extract dense features from lung abnormalities through iterative down-sampling and up-sampling.

**Table 1  Literature survey.**

| Reference | Year | Approach | Performance Measures | Advantage | Disadvantage |
|---|---|---|---|---|---|
| *Zhang et al. (2022)* | 2022 | M-SegSEUNet-CRF | Dice coefficient=85% Sensitivity= 82% | It achieves good results on smaller pulmonary tumors. | Need to improve accuracy |
| *Lakshmanaprabu et al. (2019)* | 2019 | ODNN-LDA | Sensitivity = 96.2%, Specificity = 94.2% Accuracy = 94.56% | It reduces the complexity of the network | Lack of transparency |
| *Jena, George & Ponraj (2021)* | 2021 | DGMM-RBCNN | Accuracy=87.79%, Sensitivity=86.02% Specificity=88.24% F1-Score=87.32% | It demonstrates an improved and optimum trade-off. | Handcrafted features |
| *Naseer et al. (2023)* | 2023 | Modified U-Net | Accuracy=97.98% Sensitivity=98.84% Specificity=97.47% Precision=97.53% F1-score=97.70% | It increases the classification accuracy | Limited generalizability to other datasets |
| *Cifci (2022)* | 2022 | SegChaNet | Accuracy=96.81% Sensitivity=95.92% Specificity= 96.27% | In complicated situations, it immediately delivers precise lung regions without any post-processing. | It has poor reliability and interpretability of the network model |
| *Vaiyapuri et al. (2022)* | 2022 | CHO-CADLCC | Accuracy=98.89% Precision=98.41% Recall= 98.25% Specificity=99.17% F-Score=98.29% | The diagnosis of this model is made accurately and quickly | It requires additional time for training and high complex process |

A new CAD approach for classifying lung cancer is presented in this article by *Vaiyapuri et al. (2022)*. Based on Gabor filtering, the suggested CHO-CADLCC method uses a noise reduction method to pre-process the data initially. The NASNetLarge model is also used for feature extraction from the pre-processed images. The lung nodule classification uses the CSO algorithm with the WELM model. Finally, the WELM model's parameters are optimally tuned using the CSO method, improving classification performance.

The major limitation of previous studies is shown in Table 1, including the need for improved accuracy and a need for more transparency. Due to poor image quality that interferes with the segmentation process, previous lung cancer prediction systems could not maintain accuracy. We propose a combined intelligent system ADBN-Optimized LightGBM to overcome these limitations for lung cancer segmentation and classification.

## PROPOSED METHODOLOGY

The ensuing stages comprise the predicted best lung cancer diagnosis using a CT image of the lung as an input: (1) Applying the median filter and average filter to pre-process data to remove noise; (2) histogram equalization is used for image enhancement; (3) a deep learning-based ADBM model is used for segmenting the defected lung area from its surrounds; (4) to extract lung cancer features and hybrid SOI-GRS is used for dimensionality reduction; (5) to create a LightGBM model that is optimized for categorizing lung cancer images using the successfully retrieved features; (6) the EHHO optimization

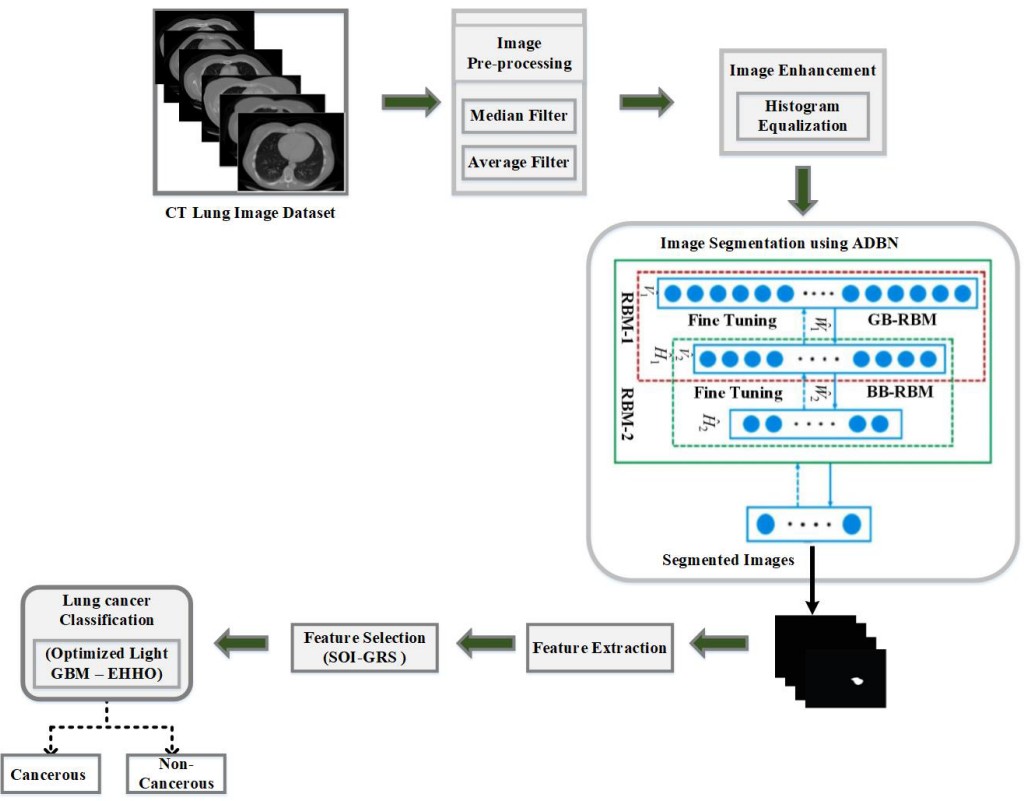

**Figure 1** The proposed combined intelligent system (ADBN-optimized LightGBM) based lung cancer detection system structure.

technique is used to optimize several lightGBM parameters for fast convergence; (7) the optimal features are identified at the end, and the lung cancer images are classified as cancerous and non-cancerous. The evaluation metrics are used to analyze the final classification performance. Figure 1 depicts the proposed work's flow diagram and highlights the steps of the established ADBN-Optimized LightGBM for an accurate lung cancer diagnosis. The input CT images are collected from the LUNA 16 dataset, the KDSB dataset, the CIA dataset, and the local dataset.

## Image pre-processing

The average filter and adaptive median filter are used in this section to eliminate excess noise from the input CT images. The image's sharpness is maintained while the median filter removes noise. In the input image, a median value of nearby pixels corresponding to each pixel is used to replace it. With rows (X) and columns (Y), a $3 \times 3$ window matrix called "M" is used in this filter. Adding zeros to two sides of an existing matrix, such as "X+2" and "Y+2," creates a new matrix, producing a size $3 \times 3$ mask. The mask is put on matrix 'M"s initial element, and the elements it points to are chosen to be sorted in ascending order. In M (1, 1), the sorted list's center element value is calculated and substituted. The

mask then moves on to the subsequent 'M' matrix element. The procedure of replacing the center element with its matching median rate is known as the value of the center element.

The spatial noise is reduced using an average filter that may arise during the input CT image's data acquisition. Each pixel's neighborhood's average value is calculated, and its associated average rates are substituted. For the image enhancement process, the noiseless image is provided by repeating the process for all elements.

## Image enhancement

The digital image quality is enhanced by using the image enhancement technique, which increases classification accuracy. Table 2 shows the pre-processing results of the proposed approach. The suggested model does this by using histogram equalization, which involves making minor adjustments to the pixel intensities.

The input digital image's histogram is calculated for histogram equalization, and then the probability distribution function is used for normalization. In the lung image, each pixel's frequency with the overall amount of pixels displayed determines the image normalization. The improved histogram transformation guarantees flatness. Below is a list of the processes and calculations involved in the procedure.

- Create a histogram using the given CT image.
- The grey levels are calculated using the Cumulative Distributive Function (CDF).
- The gray levels are computed using $CDF_1 = CDF * (\text{No. of Graylevels} - 1)$
- The image's pixels are matched to the grey levels.
- The modified histogram is plotted using this.

'm' is the discrete grayscale image, and the grey level's frequency range 'k' is represented as 'n_k.' Pixel frequency probability is computed as,

$$P_m(k) = P(m = k) = \frac{n_k}{n}, 0 \le k < N \tag{1}$$

In the lung image, the total number of gray levels is represented as 'N,' and the total number of image pixels is denoted as 'n.' The pixel rate equivalent image histogram 'k' is denoted as P_m (k) and normalized between the range of [0, 1].

$$CDF_m(k) = \sum_{j=0}^{k} P_m(k = j) \tag{2}$$

where CDF represents the cumulative distribution function

$$CDF_G(k) = kH \tag{3}$$

where the constant factor is represented as H. The transform function is processed by using the property of the cumulative distribution function, which is given as

$$CDF_G\left(G'\right) = CDF_G(T(r)) = CDF_m(r) \tag{4}$$

where [0, N] is the range of 'r' with the {m}. The resultant histogram equalization is given as,

$$G' = G.(max\{m\} - min\{m\} + min\{m\}) \tag{5}$$

**Table 2** Pre-processing results of the proposed approach.

| S. No | Input Image | Noise Removed Image | Enhanced Image |
|:---:|:---:|:---:|:---:|
| 1 |  |  |  |
| 2 |  |  |  |
| 3 |  |  |  |
| 4 |  |  |  |
| 5 |  |  |  |

## Segmentation using ADBN

The affected region is extracted from an enhanced image in the second phase, which is called segmentation. ADBN is used for segmentation in this research since it processes the image using several layers. Additionally, the network contains a significant volume of data

that has been gathered through previous analysis, which aids in accurately identifying the defective region and uses the least estimation time. This segmentation network has a more substantial impact on determining the data probability. The DBN employs a restricted Boltzmann machine (RBM) with a multi-layer concept. Understanding DBN's functioning mode may be done using the RBM operating idea. The visible and hidden layers of the RBM deep learning system are designated by *VL* and *HL*, respectively. Between visible and hidden layers, there is no relation. The particular steps of the DBNs are as follows:

First Step: The total probability distribution of the visible (*VL*) and hidden (*HL*) layers is given as

$$Prop(HL|VL) = Prop(HL|VL) = \cdots = Prop(HL|VL) \tag{6}$$

The minimized loss function is solved within the visible and hidden layers by determining the weight W.

In the second step, $(VL)_i (HL)_j$ represents the visible and hidden layer nodes; the visible layer range is denoted as I, and the hidden layer range is denoted as j for converting the RBM data from the visible layer to the hidden layer. The weight element between the visible layer node and the hidden layer node is represented by the weight value $(WT)_{ij}$.

$$\widehat{E}((VL),(HL),\varphi) = -\sum_{ij}(WT)_{ij} \cdot (VL)_i \cdot (HL)_j - \sum_i Yi \cdot (VL)_i - \sum_{ij} Xi \cdot (HL)_j \tag{7}$$

where, $\varphi = \{WT, X, Y\}$.

The above equation represents the total design combined energy; here, the RBM coefficients are represented by $\varphi = \{WT, X, Y\}$.

Third step: The joint probability distribution of the setup may be determined using the Boltzmann distribution. The value of $\Omega(\Phi)$ represents the distribution function, and $e^\star$ represents the potential function.

Where the model parameters are represented as $\varphi = \{WT, X, Y\}$, the visible unit i's binary states are represented as $(VL)_i$, the hidden unit j's binary states are represented as $(HL)_j$, their biases are represented as Xi, Yj, and the weight between visible and hidden units are represented as $(WT)_{ij}$. The network uses this energy function to give each possible pair of visible and hidden vectors probability.

$$Prop((VL),(HL),\varphi) = \frac{1}{\Omega(\Phi)} exp\{-\hat{E}((VL),(HL),\Phi)\} \tag{8}$$

Fourth step: Summarizing all potential visible and hidden vector pairings yields the partition function $(\Omega)$

$$\Omega(\Phi) = \sum_{(VL)(HL)} exp\{-\hat{E}((VL),(HL),\Phi)\} \tag{9}$$

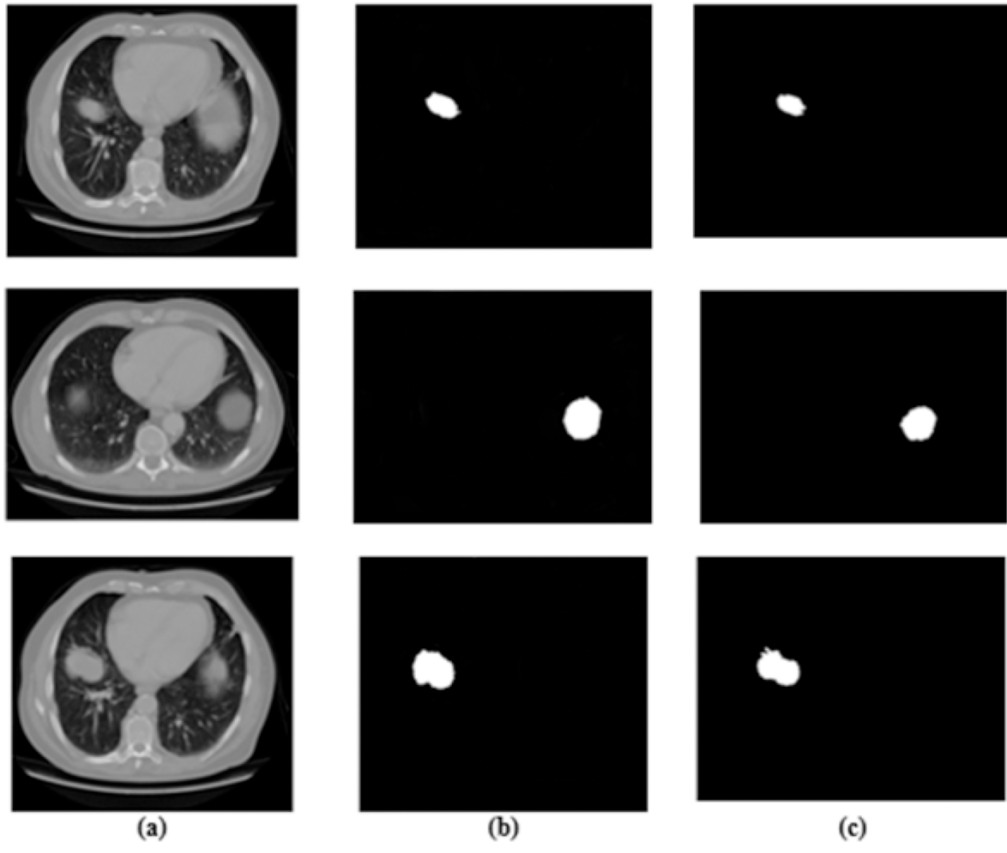

(a)          (b)          (c)

**Figure 2** **(A–C) Phases of pre-processing.**

Fifth step: With the hidden unit (*HL*) and visible vector (*VL*), the logistic functions determine the activation conditional probability:

$$Prop\big((HL)_j = 1|(VL)\big) = sigmoid\left(\sum_i (VL)_i.(WT)_{ij} + Yj\right) \tag{10}$$

$$Prop((VL)_i = 1|(HL)) = sigmoid\left(\sum_i (HL).(WT)_{ij} + Xi\right) \tag{11}$$

where, $sigmoid(k) = 1/(1 + exp(-k))$.

The affected region is successfully obtained by applying the segmentation process to the noise-removed lung image. From the input lung image, Fig. 2 demonstrates that the proposed hybrid approach properly distinguishes the affected and normal regions based on the discussion. Several characteristics are retrieved from the derived region that primarily aid in accurately predicting the abnormal region.

## Feature extraction

The extraction of significant lung features is the third phase in the proposed approach. Several features, including standard deviation, mean, variance, fourth-moment kurtosis,

and third-moment skewness, are extracted during the feature extraction process. The specific features are retrieved from the complete collected image and are recovered from a segmented region where each patient has several perceptual images (*Gudur, Sivaraman & Vimal, 2023*). The analysis has improved the image's three-dimensionality. Therefore, the dataset's dimensionality has to be decreased to improve the classification process. From the collection of features, a feature selection technique is employed for selecting the related lung features for efficient classification.

## Lung feature selection

The selection of the lung features, which is the fourth phase of the proposed research, is carried out using a hybrid SOI-GRS technique. The introduced feature selection approach has many advantages, such as simple identification of hidden patterns, the simplest optimization procedure, easy-to-understand layout, local searching, rapid selection results, minimal control variables, and simple understanding. Due to these merits, the hybrid SOI-GRS method is employed for selecting significant relevant features in this work. The spiral effect is used by the working process of the spiral optimization algorithm to choose features while solving the unconstrained optimization problem.

The feature selection process works properly. The features are effectively selected by utilizing periodic descent direction and convergence setting. By exploitation (good solution) and exploration (global solution), the approach predicts optimization characteristics with the aid of the settings. Instead of using a single gradient function when choosing an optimization process, this approach uses numerous spiral points, making it easier to determine the current optimal point. The features are processed to select the better solution from the set of points *via* the most significant number of iterations. These are the spiral optimization process's algorithmic steps, as determined by the research,

The stages of the stated algorithm compute the results, which are regarded as chosen characteristics. Utilizing the search parameters, the search point's performance has to be enhanced. The initial points, rank r(k), and rotation matrix $R(\theta)$ define the parameters. The descend direction setting must be done first based on the initial configuration.

$$R(\theta) = \begin{bmatrix} 0_{n-1}^T & -1 \\ I_{n-1} & 0_{n-1} \end{bmatrix} \tag{12}$$

The above matrix $I_{n-1}$ is estimated as follows,

$$I_{n-1} = (n-1)*(n-1). \tag{13}$$

The identity matrix is described by Eq. (13), and the zero vector is expressed as,

$$0_{n-1} = (n-1)*1 \tag{14}$$

Once defined, the initial point indicated in R must be satisfied by the below.

$$\min_{i=1,...m} \left\{ \max_{j=1,2,...m} \left\{ rank\left[ d_{j,i}(0), R(\theta) d_{j,i}(0), ...R(\theta)^{2n-1} d_{j,i}(0) \right] \right\} \right\} \tag{15}$$

$$d_{j,i}(0) = x_j(0) - x_i(0) \tag{16}$$

$$r(k) = r = \sqrt[kmax]{\delta} \tag{17}$$

$\delta > 0$ and $\delta = 1/k_{\max}$ or $\delta = 10^{-3}$.

To enhance the effectiveness of the search process, the convergence setting must be specified after specifying the descending direction setting of the searching points. This procedure is carried out up until the $k_{max}$ maximum iteration.

$$r(k) = \begin{cases} 1 & (k^* \leq k \leq k^* + 2n - 1) \\ h & (k \geq k* + 2n) \end{cases} \tag{18}$$

$$h = \sqrt[2n]{\delta}, \delta \in (0, 1) \delta = 0.5.$$

The feature selection procedure is regularly carried out in search space based on parameters mentioned above to enhance the feature selection process. From the set of features, the global solution is selected by this process. In the search space, selected features are arranged, and their effectiveness in accurately assisting in the detection of lung cancer must be evaluated using a generalized rough set approach. From the feature set, the primary advantage of using the rough set is the efficient selection of the optimized subset. Additionally, it minimizes the time required to minimize the number of undesirable features and attributes. For effectively predicting the optimized features, the upper and lower approximation approaches are used by the rough set.

The effective lung cancer characteristic is chosen using the procedure mentioned earlier, and it is then repeated until the maximum number of iterations has been reached. The next stage, described below, involves classifying the chosen characteristics with the specified features to identify normal and abnormal lung images.

## Classification

The lung cancer classification is carried out using a machine-learning classifier. The new technique improves the overall effectiveness of the suggested strategy while maximizing the accuracy of cancer prediction. An optimized Light GBM model is presented in this research using the EHHO algorithm for lung cancer classification. The approaches of gradient-based one-side sampling (GOSS) and exclusive feature bundling (EFB) solve the over-fitting issue in the traditional EHHO. For optimizing different lightGBM parameters, fast convergence is offered by the EHHO optimization algorithm. The optimized parameters in the proposed classification approach are minimal data in the leaf, maximum depth, number of training iterations, *etc*.

Compared to the existing classification models (*Shafi et al., 2022*), this one's benefit is that it takes less time and offers various financial instruments dispersed over several computers. The normalization is performed from 0 to 1 to improve performance and eliminate the impacts of varied dimensions.

$$\tilde{y} = \frac{y - min \, y}{max \, y - min \, y} \tag{19}$$

*max y* and *min y* show the maximum and minimum data values, *y* denotes the data's actual value, and $\tilde{y}$ denotes the data's normalized value. Gradient boosting decision tree (GBDT) is one of the most often used classification techniques.

In the training set $\{(y_1, z_1), (y_2, z_2), M, (y_0, z_0)\}$, $y$ specifies the data samples, $z$ specifies the class labels, and $G(y)$ represents the estimated function. The loss of functions $(Mz, G(y))$ is reduced using the GBDT approach.

$$\widehat{G} = \underset{G}{ARMIN} F_{y,z} [M(z, G(Y))] \qquad (20)$$

After that, a line search is used to obtain the GBDT iterative criterion to minimize the loss of functions.

$$G_n(y) = G_{n-1}(y) + \lambda_n i_n(y) \qquad (21)$$

The above equation $\lambda_n = \underset{\lambda}{ARMIN} \sum_{j=1}^{0} M\left(z_j, G_{n-1}(y_j) + \lambda i_n(y_j)\right)$ ( )s represent the base decision trees, and n specifies the iteration number. When the total number of samples or feature dimension is increased, the function of GBDT cannot generate effective results.

The most considerable expense in decision tree learning is the split point identification once the basis classifiers and ensemble method have been treated as the decision tree. LightGBM is a very efficient GBDT that uses GOSS and EFB. The information gain is frequently used to divide each node in GBDT. LightGBM uses GOSS for determining the split points. For splitting the points, the mathematical descriptions are displayed as follows;

$$W_k(e) = \frac{1}{o} \left[ \frac{\left( \sum_{y_j \in B_m} h_j + \frac{1-b}{c} \sum_{y_j \in B_m} h_j \right)^2}{o_m^k(e)} + \frac{\left( \sum_{y_j \in B_s} h_{j-} + \frac{1-b}{c} \sum_{y_j \in C_s} h_j \right)^2}{o_s^k(e)} \right] \qquad (22)$$

The above equation $B_m = \{y_j \in B : y_{jk} \le e\}$ $B_s = \{y_j \in B : y_{jk} > e\}$ $C_j = \{y_j \in C : y_{jk} \le e\}$ $C_s = \{y_j \in C : y_{jk} > e\}$ $h_j$ represents the negative gradients of the loss function, and $\frac{1-b}{c}$ is used for gradient normalization.

LightGBM has combined the unique features. The feature scanning method's construction of similar feature histograms using feature bundles is modeled. $P(D_a * F_e)$ Reduces the LightGBM computational complexity. LightGBM implements the GBDT effectively using GOSS and EFB to improve computation efficiency. Computing the variance gain allows the GOSS to divide the best node. The GBDT's training process increases using the EFB by grouping multiple unique features into less dense features. The weighted combinations approach is used to get the LightGBM design $G_N(y)$. $i_n(y)$ denotes the base decision trees, and N denotes the maximum number of iterations.

### EHHO algorithm for parameter optimization

Initially, the light GBM model trains images from a lung cancer dataset. When the data dimensions are increased, the prediction result's performance is decreased. The optimized classification approach infuses the EFB and GOSS concepts for effective classification to solve this performance degradation. EFB accelerates the training process, and the variance gain is used for splitting the optimal nodes through GOSS. The computational effectiveness is enhanced without affecting prediction accuracy, and the optimized light GBM model reduces the loss function. For classification, the EHHO ignores ineffective

losses and further improves system performance. Until it finds a perfect solution, the EHHO algorithm iteratively appears for better solutions.

EHHO is represented by a gradient-free optimization algorithm based on population; the searching capability of prey is used to simulate this algorithm, and this algorithm uses various pursuit techniques. The convergence and multi-objective optimization problems are all resolved by the EHHO method. In the EHHO algorithm, the capability of exploitation is enhanced by introducing the elite fractional mutation approach further to improve the efficiency and accuracy of an algorithm. Like other optimization techniques, the EHHO algorithm randomly initializes the population members.

$$X^{(0)} = \left( P_1^{(0)}, P_2^{(0)}, P_3^{(0)}, \ldots, P_n^{(0)} \right). \tag{23}$$

Here, the total hawk population is represented as n; $y^{th}$ population individual is given by $P_y^{(0)} = \left\{ P_{y1}^{(0)}, P_{y2}^{(0)}, \ldots, P_{yh}^{(0)} \right\}$, where the decision variable dimension is represented by h. Parameters are optimized in the EHHO algorithm using the two significant steps: exploration and exploitation.

The minimum data in the leaf (MDL), the number of training iterations (NTI), the number of leaves (NL), and the maximum depth (MD) are significant parameters that are optimized for reducing the lower complexity of the model. Over-fitting occurs as the value of ML rises, and the same issue also arises when there are too many NTIs. MDL, NTI, NL, MD, and are initially set to the values (20, 40... 250), (10, 20... 450), (20-256), and (1, 2... 12). Table 3 displays the optimized parameters of the classification network.

## Performance evaluation and simulation

We performed much research utilizing chest CT scans as a test platform to determine if they related to normal or lung cancer patients. The datasets, performance metrics, model hyper-parameters, results, and performance comparison are described in this section.

## Dataset description

Four different datasets are used for the evaluation of the proposed approach. Training and testing were conducted using LUNA 16, KDSB, CIA, and local datasets. CIA collection's CT lung images are used to diagnose or predict the presence of lung cancer.

In the DICOM file format, the data was recorded with particular tags for imaging, research dates, birth dates, and other information. The imaging modalities obtained include CT, PT, DX, and CR. Among different patients, 5,043 images were used to assess lung cancer. A collection of CT scan images in DICOM format, LUNA 16, and KDSB datasets are also included. There are 2,101 labeled data points in the KDSB dataset. The numerals 0 and 1 are used to represent it. A regular or cancer-free (negative result) is signified by the label 0, and abnormal or with cancer (favorable outcome) is reflected by the label 1. From the local dataset, the lung cancer-associated modalities are assessed using 8,700 images as testing images. Figure 3 displays the bar plot for each category for distributing images gathered from various sources.

For predicting the changes in the lung cells, the acquired radiographic CT lung images are examined by using medical methodologies. A total of 15,414 images in the entire dataset

| Table 3 LightGBM's optimized parameters. | |
|---|---|
| **Parameters** | **Optimal value** |
| MDL | 40 |
| NTI | 200 |
| NL | 230 |
| MD | 10 |

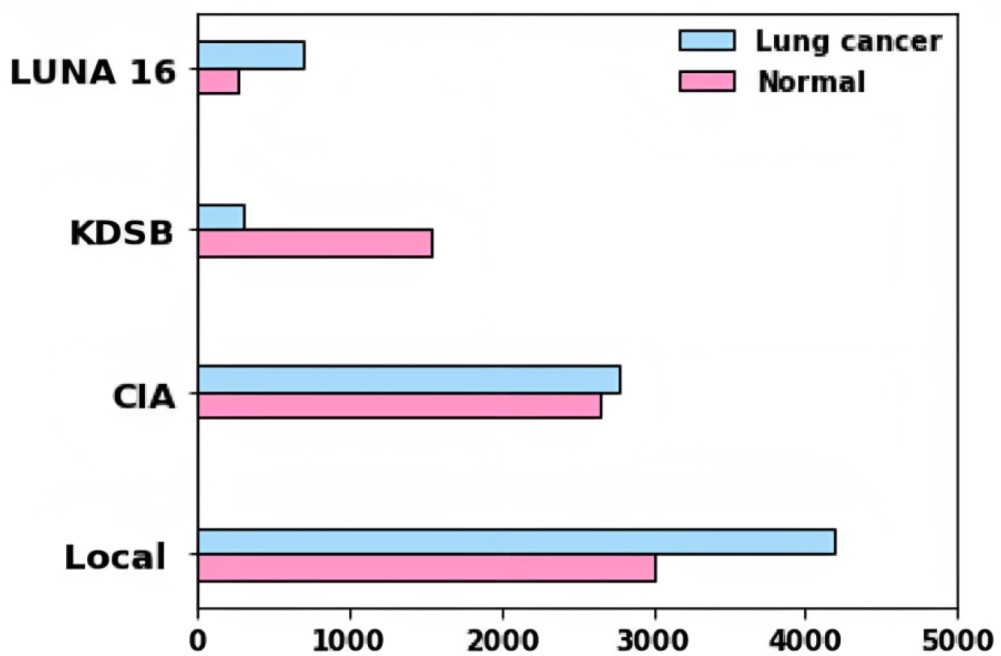

**Figure 3** Results for segmentation: (A) lung image, (B) ground truth, and (C) segmented image.

were for experiment analysis. In total images, 7,954 images are cancer-labeled images, and 7,460 images are normal. Numerous examples of lung nodules ranging in size from a few millimeters to several centimeters are seen in our dataset. The tumors in this collection include nonsolid nodules, partially solid nodules, and solid nodules, representing various pathological types.

To conduct and evaluate our experiment, we randomly divided the medical image collection into three sets: training, validation, and testing. We could generalize and examine the ADBN-Optimized LightGBM model using data from a larger dataset due to the dispersion of the training, validation, and testing sets from different databases. Table 4 displays the distribution of images with two classes for each train, validation, and test set category.

According to 'Related Prior Works', a small, imbalanced dataset was the foundation for most of the research. We were able to create an effective deep-learning model. This proposed effective deep learning model balances the variance trade-off since we had a

**Table 4  Dataset distribution.**

| Category | Training set | Validation set | Test set |
|---|---|---|---|
| Non-lung cancer | 2,891 | 2,166 | 2,403 |
| Lung cancer | 2,145 | 2,845 | 2,955 |

more significant and varied dataset. The learning rate, batch size, number of iterations, and other important criteria are adjusted using the validation set. The evaluation of the ADBN-Optimized LightGBM model subsequently used the test set, which possesses an entirely unknown dataset.

## Evaluation metrics

The affected lung cancer region is successfully segmented using many layers of a segmentation network. Because the segmented region is typically utilized to obtain the most valuable features, its accuracy must be assessed.

Metrics, including F1-score, specificity, accuracy, and sensitivity, ROC curve, are used to calculate the performance of the proposed lung cancer diagnosis system. False negative (FN), true negative (TN), false positive (FP), and true positive (TP) measures are used to analyze these metrics. TP measures the number of correctly identified cancer-infected malignant lung predictions. The number of correctly predicted non-cancerous lung as uninfected cases is known as TN. FP represents the amount of inaccurate predictions in identifying a healthy lung with cancer. FN is the number of times a cancer-infected lung was mistaken for a healthy lung.

$$Accuracy = \frac{TP + TN}{TP + FP + TN + FN} \tag{24}$$

$$Sensitivity = \frac{TP}{TP + FN} \tag{25}$$

$$Specificity = \frac{TN}{TN + FP} \tag{26}$$

$$Precision = \frac{TP}{TP + FP} \tag{27}$$

$$Recall = \frac{TP}{TP + FN} \tag{28}$$

$$F - measure = 2 * \frac{Precision * Recall}{Precision + recall} \tag{29}$$

The receiver operating characteristic (ROC) curve is used to calculate the Classification accuracy. TPR and FPR are generally used for plotting the curve at various threshold levels. Recall or sensitivity determines TPR, whereas fallout is used to establish FPR. The fallout and sensitivity of the classifier are plotted along a ROC curve. One of the effects of known specificity is shown as fallout.

## Model hyper-parameters

The best SGD method, the Adam optimizer, was chosen because it combines the most outstanding qualities of RMSProp and AdaGrad. During training, it can handle noise and

sparse gradients. Reaching the global minimum is quicker and more effective when the default hyper-parameters are used. The Adam optimizer optimizes the hyperparameter of the deep learning-based ADBN segmentation network. The machine learning-based LightGBM classifier is optimized using the EHHO parameter optimization technique for classification.

The Adam optimizer (*Shankar et al., 2022*) is selected for updating the hyper-parameters of the ADBN for training. The updated parameters are given in Table 5. The momentum helps accelerate convergence by adding a fraction of the past update to the current update. We optimize the momentum moderate value with 0.9 because a moderate momentum helps the model overcome small gradients and speed up convergence. We used the learning rate of 0.001 for network training; an optimal learning rate ensures stable convergence without leading to numerical instability. It helps the model learn the underlying patterns in the training data without memorizing noise, leading to a more robust and accurate model.

Training deep neural networks on large datasets can be computationally intensive. Using a smaller number of epochs of 10 can be advantageous regarding computational efficiency, as it reduces the overall training time. Limited computational resources, such as GPU availability, may lead to the need for shorter training times. Using a minimum number of epochs, the network can conserve resources while obtaining reasonable model performance. A minimum number of epochs allows early stopping if the model converges quickly.

To create and implement the deep learning systems, we used Python 3.6 for the deep learning models. Additionally, metrics for performance, including F1-score, confusion matrix, precision, accuracy, and recall, are computed using the Scikit-Learn Python module.

## EXPERIMENTAL RESULTS

A training and testing phase utilizing the gathered dataset was conducted simultaneously with constructing the ADBN-Optimized LightGBM. Different datasets were used to train and test the binary classification models individually. The categorization assessment measures are displayed first after the model has been adapted, then superior classification results utilizing the ADBN-Optimized LightGBM. The performance of each result in the test set is used to evaluate it.

We employed the binary classification approach to differentiate between chest CT images of patients with lung cancer and healthy individuals. Following the implementation of our binary classification model, we observed an improvement in all assessment metrics. The performance metrics for our model's training, validation, and testing are summarized in Tables 6, 7 and 8. For binary classification, the proposed (ADBN-Optimized LightGBM) model's overall training and validation accuracy is 99.7% and 99.53%, respectively, with a test set accuracy of 99.87%. For both non-lung cancer detection and lung cancer detection, the proposed model achieves 99.01% and 98.86% F1 scores, respectively. The model identified five false-positive and four false-negative predictions. Our model achieved a 99.52% binary classification precision. The proposed approach for binary

**Table 5   Optimized parameters of the network.**

| Hyper-parameters | Range |
|---|---|
| Initial learning rate | 0.001 |
| Mini-batch size | 64 |
| Maximum number of epochs | 10 |
| momentum | 0.9 |
| Gradient decay factor | 0.9000 |

**Table 6   Performance results of the proposed lung cancer classification system for training.**

| | Accuracy (%) | F1-Score (%) | Recall (%) | Precision (%) |
|---|---|---|---|---|
| Lung cancer | 99.7 | 99.84 | 99.65 | 99.3 |
| Non-cancer | | 99.53 | 99.23 | 99.02 |

**Table 7   Performance results of the proposed lung cancer classification system for validation.**

| | Accuracy (%) | F1-Score (%) | Recall (%) | Precision (%) |
|---|---|---|---|---|
| Lung cancer | 99.53 | 98.89 | 99.51 | 99.89 |
| Non-cancer | | 99.03 | 98.8 | 98.97 |

classification achieves the most incredible accuracy and F1 score. It demonstrates the excellent classification and generalization capabilities of our model.

The accuracy and losses for training and validation for binary classification are shown in Figs. 4 and 5. Large fluctuations may be seen in the accuracy and loss graph to validate our approach. Around the decision boundary, the model's training parameters are slightly modified during training. The validation accuracy is occasionally more impacted than the training accuracy, leading to substantial variation and volatility. The training accuracy is improved by making minor adjustments to the training parameters. The validation set's smaller size causes this than the training set.

In contrast, the infrequent variations in validation accuracy do not impact the model's performance. The proposed network takes 10 epochs for training, allowing quicker training than running the model for a more significant number of epochs. The over-fitting problem of the network is avoided by using a smaller number of epochs because our proposed hybrid deep learning model is complex and has many parameters.

The ROC shown in Fig. 6 further demonstrates the effectiveness of the suggested method for classifying lung cancer. The above result demonstrates that segmentation (ADBN) and classification (Optimized LightGBM) models improve the proposed intelligent system's classification efficiency.

The ADBN-optimized LightGBM model's test set confusion matrix for binary classification is also shown in Fig. 7. In addition, 5,358 sample patches are split into two categories: cancer and non-cancer. To assess the effectiveness of the ADBN-Optimized LightGBM model, a total of 2,403 patches are employed in the non-cancer group, and it
**Table 8** Performance results of the proposed lung cancer classification system for testing.

|  | Accuracy (%) | F1-Score (%) | Recall (%) | Precision (%) |
|---|---|---|---|---|
| Lung cancer | 99.87 | 99.01 | 99.75 | 99.56 |
| Non-cancer |  | 98.86 | 99.30 | 99.42 |

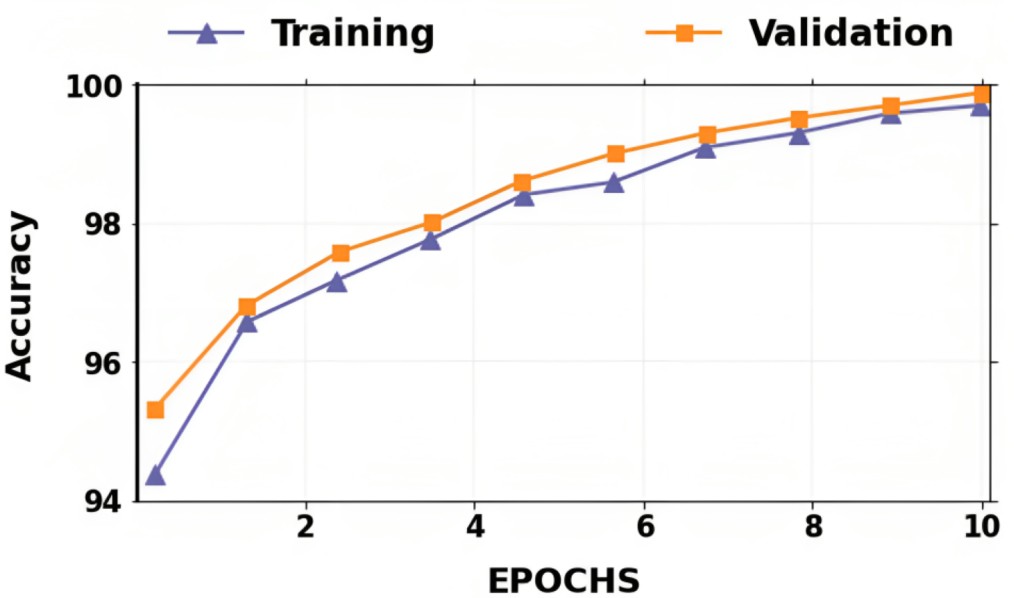

**Figure 4** Dataset distribution.

accurately identifies 2,399 sample patches as non-cancer and incorrectly identifies four sample patches. For cancer prediction, 2,955 sample patches are utilized in the cancer group; the ADBN-Optimized LightGBM model incorrectly classifies five sample patches as non-cancerous, whereas 2,950 sample patches are classified as cancerous.

## Ablation experiments

An ablation experiment is carried out in this section to illustrate the impact of the proposed approach. There are five modules in the proposed approach. They are pre-processing, ADBN-based segmentation, feature extraction, SOI-GRS-based feature selection, Light GBM-based classification, and EHHO-based parameter optimization modules. Our experimental results show that the proposed approach may attain outstanding performance. We carried out the ablation experiment in this section to investigate the impacts of each module in the suggested architecture.

LUNA 16, KDSB, CIA, and local datasets for classifying normal and abnormal lung cancer are among the combination datasets utilized for experiment analysis. This was accomplished by applying quantitative findings to networks with different module combinations, as indicated in Table 9. With the feature selection step, the dataset's F1-score, recall, precision,

 

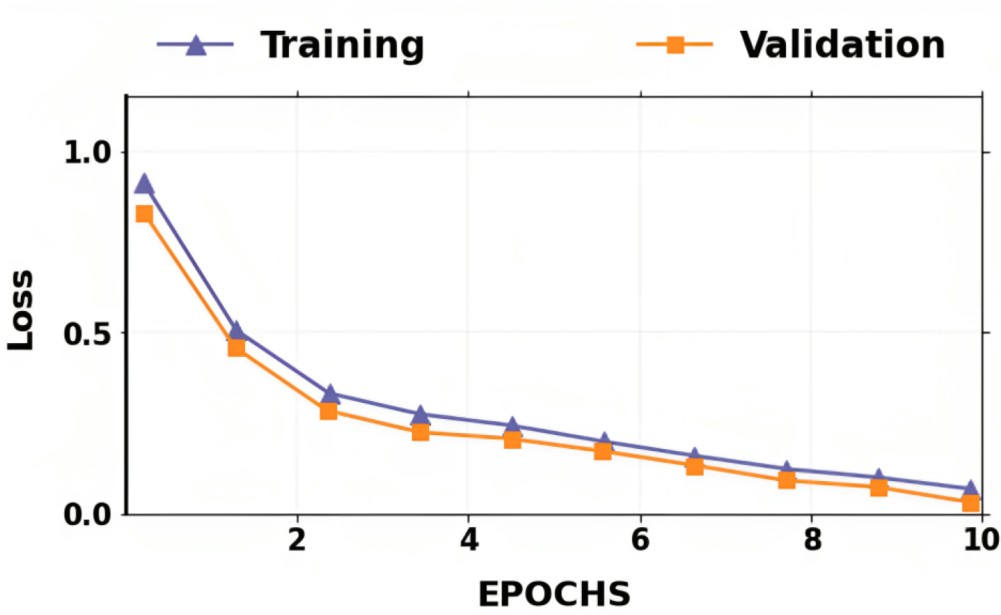

**Figure 5   Accuracy of training and validation for the proposed lung cancer classification system.**

**Table 9   Ablation experiments on each module.**

| Pre-processing | Segmentation | Feature extraction | Feature selection | Classification | Hyperparameter optimization | Accuracy (%) |
|---|---|---|---|---|---|---|
| ✓ | ✓ | ✓ |  | ✓ | ✓ | 91.23 |
|  | ✓ | ✓ | ✓ | ✓ | ✓ | 90.23 |
|  |  | ✓ | ✓ | ✓ | ✓ | 85.06 |
| ✓ | ✓ | ✓ | ✓ | ✓ |  | 97.02 |
| ✓ | ✓ | ✓ | ✓ | ✓ | ✓ | 99.87 |

and accuracy metrics show much higher results, and the data classification process takes longer. Additionally, the lack of the pre-processing phase has a minor effect on the final results. Classification accuracy is slightly lower when hyperparameter optimization is not used instead of parameter optimization.

However, without SOI-GRS-based feature selection, it only attains 91.23% accuracy. Additionally, even without parameter optimization, it attains 97.02% accuracy. The absence of pre-processing and segmentation significantly impacts the effectiveness of classification. Likewise, the suggested method produces excellent results using all modules, including pre-processing, segmentation, feature extraction, feature selection, classification, and parameter optimization. Since the modified ADBN's deep supervision capacity enables it to achieve better segmentation results than regular 2D and 3D networks, this aids the suggested framework in achieving successful image detection results. In the end, it is found that improved classification outcomes are obtained by combining optimized LightGBM-based classification with ADBN-based segmentation and pre-processing.

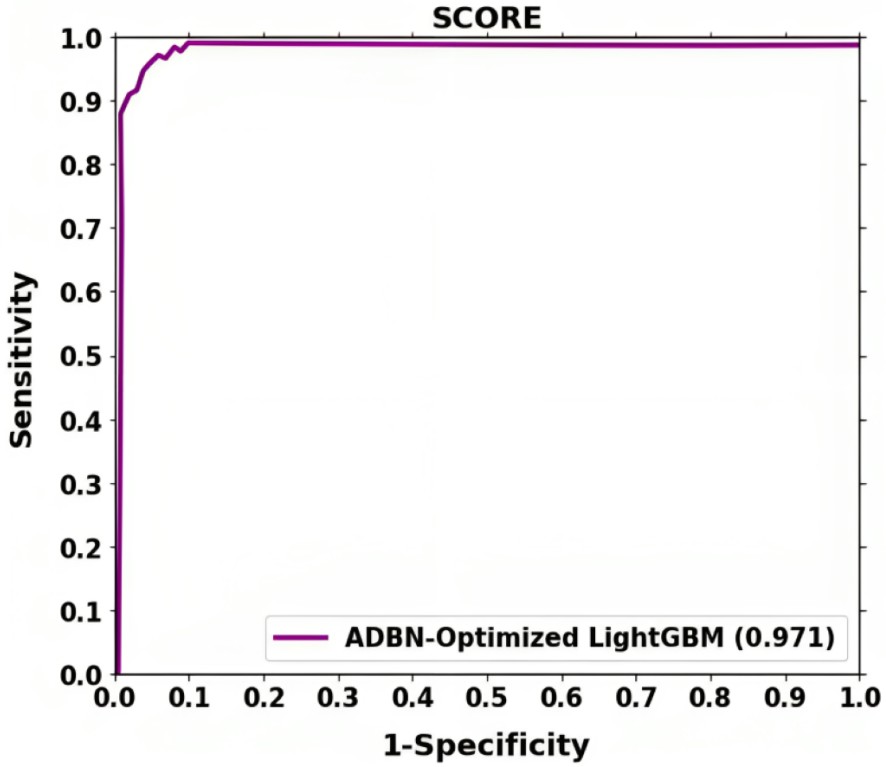

**Figure 6  Training loss and validation loss for the proposed lung cancer classification system.**

## Performance comparison

This section compares the classification outcomes of our suggested approach to those from earlier studies using CT scans to identify lung cancer in terms of performance measures such as accuracy and precision. The most cutting-edge classifiers for lung cancer classification, listed in Table 10, are used to thoroughly examine the proposed ADBN-optimized LightGBM model in terms of accuracy and precision. Several performance measures are used in the existing approaches, but we have taken two, such as accuracy and precision, for comparison analysis.

Lung cancer detection and classification with the DGMM-RBCNN technique was developed by *Jena, George & Ponraj (2021)*. The LIDC dataset is thought to be employed in this investigation to deal with the more significant amount of scanned images. Image noise is reduced by applying the Wiener and Gaussian filters. Third, region-growing segmentation is employed to obtain precise region of interest (ROI) segmentation. Next, features like intensity, entropy, perimeter, and area statistically based features are retrieved that are highly important for a nodule of interest. A deep Gaussian mixture model (DGMM) decreases dimensionality from these collected features. Finally, a region-based convolutional neural network (RBCNN) classifies lung cancer as normal or abnormal. This model produces an 87.79% accuracy and 89% precision. Compared to all existing models, this model produces less accuracy and precision below 90%.

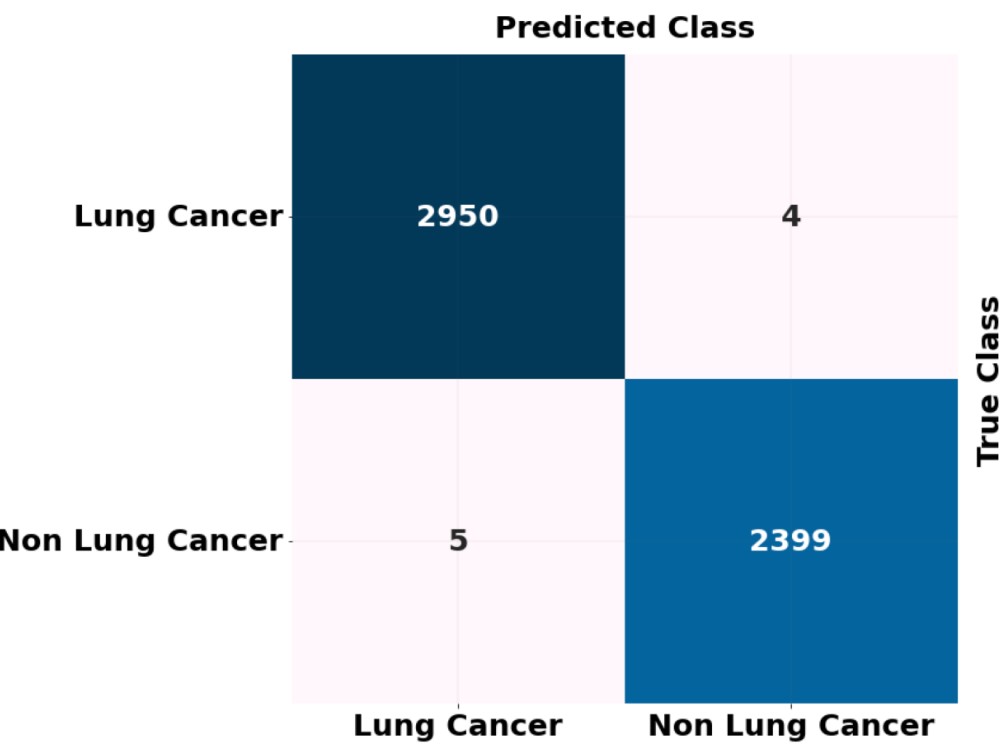

**Figure 7** ROC curve attained for proposed lung cancer classification system.

**Table 10** Performance analysis of proposed approach and cutting-edge approaches on lung cancer classification on the CT image dataset.

| Reference | Year | Method | Dataset | Accuracy (%) | Precision (%) |
|---|---|---|---|---|---|
| *Jena, George & Ponraj (2021)* | 2021 | DGMM-RBCNN. | LIDC | 87.79 | 89 |
| *Naseer et al. (2023)* | 2023 | Modified AlexNet -SVM | LUAN16 | 97.98 | 97.53 |
| *Jagadeesh & Rajendran (2023)* | 2023 | PNN | LIDC-IDRI | 95.88 | - |
| *Cifci (2022)* | 2022 | SegChaNet | External source | 96.81 | 92.15 |
| *Khan et al. (2021)* | 2021 | VGG-SegNet | LIDC-IDRI and Lung-PET-CT-Dx | 97.83 | 97.05 |
| *Nanglia et al. (2021)* | 2021 | FFBPNN-SVM | ELCAP | 98.08 | 98.17 |
| *Huidrom, Chanu & Singh (2022)* | 2022 | FFNN | LIDC | 95.5 | - |
| *Chilakala & Kishore (2021)* | 2021 | DBNGHHB | Resources Initiative database | 97.52 | - |
| *Aswathy et al. (2022)* | 2022 | BoVW–CRNN | LIDC | 99.35 | 96.5 |
| Proposed classification system | | ADBN-Optimized LightGBM | LIDC-IDRI | 99.87 | 99.52 |

Utilizing computational intelligence approaches, lung cancer is precisely identified and categorized using CT images via the development of an automated system by *Naseer et al. (2023)*. Lobe segmentation, potential nodule extraction, and nodule classification as either cancerous or non-cancerous are usually steps in this process. The developed lung cancer classification uses a modified U-Net-based lobe segmentation and nodule detection model

with three stages. The lobe is segmented in the first phase using a CT slice and predicted mask, while the predicted mask and label are used to extract the candidate nodule using a modified U-Net architecture in the second phase. The candidate nodules are classified as cancerous or non-cancerous using an SVM classifier based on a modified version of AlexNet. LUAN16 dataset is used for implementation analysis. The modified AlexNet-SVM classification model classifies lung cancer with 97.53% precision and 97.98% accuracy.

For segmenting and classifying lung cancer, an enhanced model was created by *Jagadeesh & Rajendran (2023)* using a genetic algorithm. The adaptive median and average filters are pre-processed filters in the model that are applied to the input CT images. The guaranteed convergence, particle swarm optimization approach separates the cancer tissues into ROIs after the filtered images have been improved using histogram equalization. The images are classified using probabilistic neural networks (PNN). The LIDC-IDRI benchmark dataset is used for experimentation. This PNN classification model achieves 95.88% accuracy in classifying normal and pathological images.

*Khan et al. (2021)* created pre-trained DL-based classification and VGG-SegNet-enabled nodule mining to enable automated lung nodule identification. The handcrafted features, including GLCM and LBP, are concatenated to improve disease detection accuracy. The deep features obtained are used to classify lung CT images. The Lung-PET-CT-Dx and LIDC-IDRI databases provide the images utilized in the research. According to the experimental findings, concatenated deep and handcrafted features in the VGG19 architecture may obtain a precision of 97.05% and an accuracy of 97.83% when used with the SVM-RBF classifier.

An SVM and neural network hybrid approach for classifying lung cancer was created by *Nanglia et al. (2021)*. Combining SVM with the Feed-Forward Back Propagation Neural Network produces a hybrid method that lowers the classification's computational complexity. The classification is performed using a three-block mechanism. The dataset is pre-processed in the first block. The features are extracted using the SURF technique in the second block. Finally, the FFBPNN model is used for classification. The accuracy of FFBPNN-SVM is 98.08% for classification.

A novel CAD system utilizing a neuro-evolutionary method was introduced by *Huidrom, Chanu & Singh (2022)*. Pulmonary nodules are identified within the lung areas the CAD system extracts from CT images. A novel neuro-evolutionary technique called feed-forward neural network (FFNN) combined with particle swarm optimization and the cuckoo search algorithm is used to reduce false positives. This method's performance is further enhanced by utilizing regularized discriminant features until it reaches 95.5% accuracy.

A deep belief network (DBN) in conjunction with an opposition-based hybrid grasshopper and honey bee optimization algorithm (GHHB) was developed by *Chilakala & Kishore (2021)* to classify lung cancer. Pre-processing procedures first increase the quality of the image, after which shape, color, and texture are retrieved. Multivariate techniques such as local tangent space alignment are employed to eliminate redundant and inactive features from feature selection methods. The CT images are classified as malignant or benign using DBN, which is calibrated to the opposition-based GHHB algorithm for the

diagnosis of lung cancer. The DBN network represents 97.52% accuracy, according to experimental findings.

A novel model was created by *Aswathy et al. (2022)*; it focuses primarily on the complex and diverse diagnosis of lung cancer, a classic, exciting, and dangerous area of oncology. A modified color-based histogram equalization and a Gabor filter were applied to improve the nano-image input. Next, the guaranteed convergence particle swarm optimization (GCPSO) technique was used to segment the lung cancer image. A nano-measuring instrument with a graphical user interface was created to categorize the cancerous area. A convolutional recurrent neural network (CRNN) and a Bag of Visual Words (BoVW) were used for feature extraction and image classification. This model achieved an average accuracy of 96.5%.

The proposed ADBN-Optimized LightGBM model outperforms existing deep learning models in terms of performance, successfully segments the defected tumor region, and categorizes the CT images into cancerous and non-cancerous regions. As a result of the classification and segmentation process, the primary advantage of the proposed ADBN-Optimized LightGBM model is that it avoids over-fitting and has no detrimental impact on network performance. The suggested method needed significantly less time than existing methods for segmenting images generally.

The proposed classification system achieves a higher classification rate than the existing systems for lung tumor classification, and it is the most efficient system for providing binary classification results based on CT scans with the most extensive collection of images ever used. The proposed framework effectively extracts features corresponding to the diseases' inter-scale heterogeneity, enhancing classification performance. The proposed hybrid approach to lung cancer segmentation and classification for classifying CT images outperforms the existing methods; the proposed method achieves a 99.87% accuracy, 99.52% precision, 99.95% recall, and 99.01% F1-Score. The proposed model automatically diagnoses and pre-screens the lung tumor due to its encouraging classification accuracy.

## Discussion

The improved average and adaptive median filters are used for noise removal, and histogram equalization is used to enhance the filtered images to improve the lung image quality. From the segmented lung image, 50 features are collected for the analysis, with each approach choosing an appropriate level of features to reduce computing costs. Using many spiral centers and settings, a total of 14 features are chosen by the hybrid SOI-GRS. Additionally, estimate standards are used for predicting the most suitable features. The introduced feature selection approach selects fewer features than recent feature selection algorithms like GAWA, PSOMS, and ACO. The overall cancer classification process is enhanced by using the hybrid SOI-GRS approach because of fewer relevant features. The reduced feature selection process parameters result in less image segmentation by ADBN. From CT images, the performance ability for segmenting defective regions is also enhanced by the ADBN with the replacement of RBMs to GB-RBM and BB-RBM. The 'Related Work' section also discusses the approaches and their drawbacks.

LUNA 16, KDSB, CIA, and the local datasets are used for the experiment analysis of the proposed approach. For all datasets, the image's size is $240 \times 240$ pixels. Results from different CT imaging datasets are compared simultaneously in the results section. Lung CT scans are classified into cancerous and non-cancerous using the Optimized LightGBM classifier. Using the EHHO technique, the LightGBM classifier's hyper-parameters are optimized. Combining CT image datasets yields the best classification outcomes using optimized LightGBM and ADBN. We employ the ReLU activation function in ADBN and the Adam optimizer for optimization. The simulation component of the dataset has three sections: training, testing, and validation. Based on various factors, we observe a wide range of results for all datasets in 'Experimental Results'.

We employ a LightGBM classifier optimized with ADBN to discover the optimal outcomes in the classification stage. The LightGBM classifier performs better than previous classifiers with ADBN and an EHHO hyperparameter optimization technique in every aspect. Comparing the suggested model with existing approaches, such as DGMM-RBCNN, FFBPNN-SVM, VGG-SegNet, and FFNN, demonstrates a superior trade-off over the prevailing techniques. The specific parameters such as F1-score, recall, precision, and accuracy improved by 1.06%, 1.293%, 2.384%, and 1.06% in the combination of the adapted DBN and Optimized LightGBM.

## Advantages and limitations
### *Advantages*

For finding the defective region of the lung, a new ADBN-based technique is proposed. The proposed system works thus effectively for two key reasons. First, modified DBN is highly suited for medical image processing because it can segment disease regions with fewer parameters and is less complicated. With less processing time and complexity, the ADBN networks are hence preferred for segmentation. In the RBM layers, including GB-RBM and BB-RBM, the over-fitting or under-fitting problems of the segmentation network are prevented by using the ReLU activation function because the over-fitting or under-fitting problems lower the error rate.

### *Limitations*

Our proposed approach has a few difficulties. A rise in sensitivity at high FP rates frequently follows a drop at low FP rates. The F1-score affects the model's NPV since it decreases when FN rates increase. Second, retrieval and prediction are determined by the relatively low information richness of the gallery set. As radiologists continue to learn about new cases, the database should include information; this area may be investigated in the future.

## CONCLUSION AND FUTURE WORKS

For effective lung tumor segmentation and classification from CT images, we have proposed ADBN-Optimized LightGBM architecture in this research. LUNA 16, KDSB, CIA, and local datasets are used for collecting the images for experiment analysis. The dataset images are divided into training, testing, and validation. After data collection, the noise is removed, and the image quality is enhanced using effective filters and histogram equalization techniques.

The affected region of the lung image is then segmented using many layers of a network to examine each pixel. To identify cancer, the segmented region is thoroughly analyzed, and several characteristics that are huge in dimension are also extracted. Spiral settings and an approximation approach are used to select optimized features, decreasing the system's dimensionality correctly. The characteristics are enhanced with the use of Optimized LightGBM, which accurately distinguishes between abnormal and normal cancer images. For lung tumor segmentation, the developed ADBN network helps us improve the results of the abnormal lung, where the angles and shapes of the affected tumor region could be detected better. Utilizing the results of experiments, the system's effectiveness is assessed, and it has the highest level of accuracy when identifying cancer. The output results show the distinction between normal and abnormal images with high accuracy, precision, recall, and F1-score with a considerable decrease in time taken for detection and epochs. The main advantage of this automated detection process is that it produced the best accuracy of 99.87% with negligible testing loss using a combined intelligent system. It is confirmed that the proposed hybrid ADBN-Optimized LightGBM model is outperforming the conventional models in lung cancer segmentation and classification.

We may eventually expand our present model to establish the patient's cancer status and the specific location of the tumor areas. For the initial lung segmentation, Watershed segmentation can be used in the future. A network expansion and more thorough hyperparameter tweaking are two more areas for development. Our models may also be expanded to create 3D cancer-related images.

### Funding
The authors received no funding for this work.

### Competing Interests
The authors declare that they have no competing interests.

### Author Contributions
- Maheswari Sivakumar performed the experiments, performed the computation work, authored or reviewed drafts of the article, and approved the final draft.
- Sundar Chinnasamy conceived and designed the experiments, authored or reviewed drafts of the article, and approved the final draft.
- Thanabal MS analyzed the data, prepared figures and/or tables, authored or reviewed drafts of the article, and approved the final draft.

### Data Availability
The data is available at figshare: Sivakumar, Maheswari; Chinnasamy, Sundar; M S, Thanabal (2023). Classification of Lung Cancer Computed Tomography Images. figshare. Dataset. https://doi.org/10.6084/m9.figshare.23741562.v3.

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
