# Peer review of "An efficient combined intelligent system for segmentation and classification of lung cancer computed tomography images"

_PeerJ Computer Science, doi:10.7717/peerj-cs.1802_

## Round 0.1 · original submission · Major Revisions

The reviewers have substantial concerns about this manuscript. The authors should provide point-to-point responses to address all the concerns and provide a revised manuscript with the revised parts being marked in different color.

**Language Note:** The review process has identified that the English language must be improved. PeerJ can provide language editing services - please contact us at copyediting@peerj.com for pricing (be sure to provide your manuscript number and title). Alternatively, you should make your own arrangements to improve the language quality and provide details in your response letter. – PeerJ Staff

Reviewer 1 ·

Basic reporting

The manuscript titled "An efficient combined intelligent system for segmentation and classification of lung cancer computed tomography images" employed an advanced segmentation technique using an Adapted Deep Belief Network (ADBN) and classified lung tumors with an optimized LightGBM model. The research achieved an impressive accuracy of 99.87% across various datasets. However, the manuscript's presentation is not well-articulated, and some sentences are ambiguous, leading to potential misunderstandings. Therefore, I recommend the authors conduct a comprehensive revision for clarity and coherence to ensure the content is clear and easily understood by readers.

Experimental design

Model Hyperparameters: While the Adam optimizer has been chosen, there's a lack of further justification on why these particular hyperparameters are deemed optimal. It would be beneficial to provide the hyperparameter tuning process or references for better evaluation of the model.

Early High Accuracy: An explanation is warranted for achieving 95% accuracy in the first epoch. This could be due to model initialization, data leakage, or other reasons. Addressing this phenomenon is pivotal for understanding the model's behavior and reliability.

Evaluation Metrics: Although a variety of evaluation metrics are employed in the paper, given potential data imbalances, it would be wise to place greater emphasis on metrics such as F1 score, recall, and specificity, rather than accuracy alone.

Model Complexity: Given the deep nature of modern neural networks, training on a relatively large dataset (15,000 images) is expected. However, depending on the depth and architecture, 10 epochs might be too few to fully exploit the data's potential. Conversely, if the model reaches such high accuracy early on, it may also suggest that the model is overly complex for the problem or the data is not challenging enough.

Validity of the findings

Consistency of Results: While the reported accuracy is impressively high, in the medical domain, even minor misclassifications can have serious implications. It would be prudent for the authors to further validate the model's performance on other independent datasets.

Interpretation of Results: When describing the experimental results, the authors are encouraged to provide a more detailed rationale for why the model could achieve such performance and contextualize it against other methods.

Overfitting Considerations: While the paper alludes to avoiding overfitting and displays the model's accuracy, loss, and other metrics in the experimental results, a deeper discussion of the model's behaviour during training and validation, like learning curves, might be more convincing.

Performance Comparison: In comparing performance with other methods, ensure that the comparison is equitable and conducted on the same test datasets. Offering more details about these comparative models, such as their training and validation processes, would make the comparison more meaningful.

Additional comments

In summary, the paper presents an intriguing approach to the problem of lung cancer classification, but further details and clarifications are necessary to solidify the robustness of its experimental design and the validity of its findings.

Reviewer 2 ·

Basic reporting

The study of "An efficient combined intelligent system for segmentation and classification of lung cancer computed tomography images" proposed a combined feature extraction and image classification method for classifying the CT images with high accuracy. However, there are some items need to be revised and improved.
1) Line 174 “(6) At the end “should be (7).
Line 212, no extra brackets for ‘m’.

2) Table 1 no accuracy, sensitivity, specificity etc.
Why do you use these 5 images in Table 2?
Figure 2 is repetitive.
You can combine Figure 8 with Table 5, 6 and 7.

3) You used F measure and F-1 score. You should use one name to make it clear for readers.

Experimental design

Why do you just compare the accuracy, when you also calculate F1 score, specificity, accuracy, sensitivity and ROC curve?

Validity of the findings

Real world cancer detection from lung CT should be like data from KDSB, which is unbalanced. How’s the performance of your combined method when applied to these data like KDSB?

Reviewer 3 ·

Basic reporting

The paper introduces an innovative approach for predicting lung cancer from CT images, aiming to overcome challenges in segmentation and classification. The method involves noise reduction through filtering, enhanced lung image quality via histogram equalization, and lung region segmentation using an Adapted Deep Belief Network (ADBN) incorporating cascaded Restricted Boltzmann Machines (RBMs) with both Bernoulli-Bernoulli (BB) and Gaussian-Bernoulli (GB) structures. Relevant features are then selected using a hybrid Spiral Optimization Intelligent-Generalized Rough Set (SOI-GRS) approach. The subsequent lung cancer classification employs an optimized Light Gradient Boosting Machine (LightGBM) model enhanced by the Ensemble Harris hawk optimization (EHHO) algorithm. Extensive experiments on datasets including LUNA 16, KDSB, CIA, and local data demonstrate accurate identification of lung cancer cases with a remarkable 99.87% accuracy and minimal classification error. The proposed integrated system (ADBN-Optimized LightGBM) outperforms other models, exhibiting potential to enhance lung cancer patient diagnosis by medical professionals. But still have some limitations as fellow.

Experimental design

1. Regarding Table 8, could the author clarify the criteria used for selecting the compared methods or research? It seems that there might be additional segmentation methods and research available for comparison. Elaborating on the selection process would enhance the understanding.

2. Concerning feature selection, it's acknowledged in some studies that such processes may result in the loss of important information. Could the author share whether any ablation experiments were conducted to demonstrate the efficacy of feature selection in this research?

Validity of the findings

In Figure 5, the inclusion of epoch numbers appears to lead to improved accuracy. Could the author provide insights into the rationale behind choosing 10 as the epoch number? An explanation of this decision would offer valuable context.

Additional comments

It's advisable to enhance the quality of the figures for better clarity and interpretation. Improving the figure quality would contribute to the overall visual understanding of the presented data

---

## Round 0.2 · accepted · Accept

Reviewers are satisfied with the revisions, and I concur to recommend accepting this manuscript.

Reviewer 2 ·

Basic reporting

All my concerns have been well addressed. The manuscript is ready to be published.

Experimental design

The study has detailed and reasonable experimental design.

Validity of the findings

Results are good and clear.

Reviewer 3 ·

Basic reporting

It agrees with the requirements.

Experimental design

It agrees with the requirements.

Validity of the findings

It agrees with the requirements.